Serum neuron-specific enolase (NSE) is associated with the overall survival of colorectal cancer: a retrospective study

Peng Junwei 1
Ma Jie 2
Lu Jian 2
Ran Hailiang 1
Yuan Zhongqin 2
Zhou Hai 2
Huang Yunchao 2 huangych2001@aliyun.com
Xiao Yuanyuan 1 33225647@qq.com
1 School of Public Health, Kunming Medical University , Kunming , China
2 The Third Affiliated Hospital of Kunming Medical University , Kunming , China
Upadhyay Rohit
Electronic publication date: 2024 Nov 22
Publication date: 2024
Volume: 12
Electronic Location ID: e18617
Received 2024 Jun 3; Accepted 2024 Nov 9
Copyright: © 2024 Peng et al.
Copyright year: 2024
Copyright holder: Peng et al.
License: This is an open access article distributed under the terms of the Creative Commons Attribution License, which permits unrestricted use, distribution, reproduction and adaptation in any medium and for any purpose provided that it is properly attributed. For attribution, the original author(s), title, publication source (PeerJ) and either DOI or URL of the article must be cited.
License URL: https://creativecommons.org/licenses/by/4.0/

Keywords: Neuron-specific enolase (NSE), Colorectal adenocarcinoma (CRAD), Prognosis, Overall survival (OS), Survival analysis

Funding: Basic Research Program of Yunnan 202201AT070200 First-Class Discipline Team of Kunming Medical University 2024XKTDTS16 Scientific Research Fund Project of Yunnan Provincial Department of Education 2023Y0798; 2024Y212 This study was supported by the Basic Research Program of Yunnan (202201AT070200), First-Class Discipline Team of Kunming Medical University (2024XKTDTS16), and the Scientific Research Fund Project of Yunnan Provincial Department of Education (2023Y0798; 2024Y212). The funders had no role in study design, data collection and analysis, decision to publish, or preparation of the manuscript.

==============================
Background

Serum neuron-specific enolase (NSE) had been associated with survival of several cancers. However, its prognostic significance for colorectal cancer (CRC) has not been effectively discussed. We aimed to investigate the relationship between baseline serum NSE and the overall survival (OS) of colorectal adenocarcinoma (CRAD) patients.

Methods

A retrospective study had been conducted by including 564 histopathology confirmed CRAD patients between January 2013 and December 2018 from Yunnan Provincial Cancer hospital, China. Cox proportional hazards model was used to estimate the crude and adjusted associations between serum NSE measured at diagnosis and the OS of the patients. Restricted cubic spline (RCS) was further applied to delineate dose-response trend of the NSE-OS association.

Results

After controlling for possible confounding factors, baseline serum NSE was significantly associated with OS in CRAD: when dichotomizing by the median, patients with higher baseline serum NSE (NSE >= 12.93 ng/mL) were observed a worse prognosis (hazard ratio, HR: 1.82, 95% CI [1.30–2.55], p < 0.01). Stratified analysis by tumor stage revealed a stronger NSE-OS association in advanced CRAD patients. RCS disclosed a prominent dose-response relationship in NSE-OS association for all CRAD patients: along with the increase of baseline serum NSE, the adjusted HR of CRAD patients increased gradually. This dose-response trend is also evident in advanced stage CRAD patients, but not in early stage CRAD patients.

Conclusions

Serum NSE measured at diagnosis might be a useful prognostic indicator for CRAD, especially for advanced stage patients.

Introduction

Worldwide, cancer is the leading cause of death. GLOBOCAN 2020 estimated that there were 19.3 million new cancer cases and 10 million deaths globally, with colorectal cancer (CRC) accounted for approximately one-tenth of new cancer cases and deaths (Sung et al., 2021). Adenocarcinomas which originate from colorectal glandular epithelial cells accounted for overwhelmingly majority (95–98%) of CRC (Fleming et al., 2012). Disparities exist for the incidence of CRC: higher in men and in developed countries (Sung et al., 2021). The main established risk factors for CRC include: obesity and physical inactivity (Karahalios et al., 2016; Robsahm et al., 2013), unhealthy diet (characterized in high intake of red and processed meats, low intake of fiber, whole grains, and calcium) (Zhao et al., 2017; Norat et al., 2017), smoking (Ordóñez-Mena et al., 2018), alcohol consumption (Fedirko et al., 2011), family history of CRC (Jasperson et al., 2010), and inflammatory bowel diseases (Olén et al., 2017). In China, along with economic transition and Westernized dietary pattern and lifestyle, the incidence of CRC has been persistently increasing (Arnold et al., 2017).

CRC has been ranked the second leading cause of death among cancers (Sung et al., 2021). Although there have been remarkable improvements in the diagnosis and treatment of CRC in the past decades, the overall survival (OS) of CRC is less optimistic: currently, the 5-year OS for all stages CRC patients is 65%, the sixth lowest among all cancers (Siegel et al., 2022). The 5-year OS for CRC patients who underwent surgery in the United States was 54.79% (Zhang et al., 2020). In China, although the age-standardized 5-year relative survival rate for CRC gradually elevated from 2003 to 2015, by the year of 2018, it was still below 57% (Zeng et al., 2018). Therefore, identifying useful prognostic markers is of great clinical importance for CRC patients.

In recent years, it has been found that many serum tumor markers were closely associated with the survival of CRC: such as indicators of systemic inflammatory response (C-reactive protein; neutrophil-to-lymphocyte ratio, NLR) (Koike et al., 2008; Templeton et al., 2014); nutritional status (albumin) (Chiang et al., 2017), and tumor burden (carcinoembryonic antigen, CEA; carbohydrate antigen, CA19-9, CA24-2) (Rao et al., 2021; Tampellini et al., 2015). Lately, studies have reported serum enzymes may also be related to cancer survival, like alkaline phosphatase and cholinesterase (Xiao et al., 2019; Ran et al., 2022). At the same time, neuron-specific enolase (NSE) has attracted study interest. The highly acidic soluble brain protein 14-3-2 was first described by Moore and McGregor in 1965 and had been renamed NSE since the neuron-specific protein has exhibited enolase activity (Marangos, Zomzely-Neurath & York, 1976). NSE is a cell-specific isoform of the glycolytic enzyme enolase and a highly specific marker of neurons and peripheral neuroendocrine cells and thus is used to identify neuroendocrine cells and diagnose malignant tumors (Isgrò, Bottoni & Scatena, 2015). NSE is one of the most reliable serum markers for the diagnosis and prognosis of small-cell lung cancer (SCLC), and a meta-analysis of 16 studies showed that elevated pretreatment serum NSE predicted poorer OS for SCLC patients, with a combined HR of 1.78 (Tian et al., 2020). In addition, serum NSE has also been associated with increased risk of neuroendocrine tumors, adult neuroblastoma, melanoma, seminoma, renal cell carcinoma, Merkel cell tumor, carcinoid tumor, anaplastic and immature teratoma, and malignant pheochromocytoma (Isgrò, Bottoni & Scatena, 2015).

Endocrine differentiation can exist, and the presence of focal endocrine cells was common in colorectal adenocarcinoma (CRAD) (Shia et al., 2002; Gulubova & Vlaykova, 2008). Endocrine differentiation can affect the expression of several neuroendocrine markers, such as NSE, chromogranin A, and synaptophysin. Among these, NSE and chromogranin A have been considered the most common markers that reflect neuroendocrine differentiation in CRC (Atasoy et al., 2003). Higher level of serum NSE had been associated with poorer tumor differentiation (Baudin et al., 1998). In a recently published study, Luo et al. (2020) disclosed that NSE level was significantly associated with tumor stage, lymph node, and distant metastasis of CRC patients, suggesting that NSE could be used as a progression biomarker of CRC. Therefore, it is reasonable to suspect that serum NSE may be associated with survival outcomes of CRC. However, although the prognostic relevance of NSE had been discussed for several types of cancer, such as SCLC, non-small-cell lung cancer (NSCLC), prostate cancer (PC), small cell carcinoma of the urinary bladder (SCCB), and pancreatic neuroendocrine tumors (NETs) (Bremnes et al., 2003; Pujol et al., 2001; Yao et al., 2016; Fan et al., 2017; Naito et al., 2017), its association with CRC survival has not been effectively explored so far.

We intended to investigate the hypothetical prognostic significance of NSE in CRC in a large sample of Chinese patients retrieved by using retrospective design.

Materials and Methods

Study design

Study subjects were collected from the Third Affiliated Hospital of Kunming Medical University, which is also the provincial cancer hospital of Yunnan province in southwest China. We retrospectively identified patients who were hospitalized from January 1 2013 to December 31 2018, and were firstly diagnosed as CRC by histopathology. Considering the predominant proportion of colorectal adenocarcinoma in CRC, we only included this specific histopathological type.

Patient information was obtained from the digitalized hospital information system (HIS) which updated on daily basis from late 2012 onward. The following key variables of study relevance were extracted: gender, age at diagnosis, ethnicity, smoking history, alcohol drinking history, body mass index (BMI), chemotherapy, curative operation, clinical stage, serum blood indicators (NSE, NLR, albumin, alpha-fetoprotein, carcinoembryonic antigen, CA125, CA19-9). Survival information was ascertained by using computer-assisted telephone interviewing follow-up system built by the hospital. The deadline for follow-up was set as December 31, 2019. The study protocol was reviewed and approved by the Ethics Review Board of Kunming Medical University (Approval number: KMMU2021MEC109). Because of the retrospective nature, informed consents had been waived.

Variables and definitions

Death of the patient due to any cause was the primary outcome. Survival time was defined as the date of the patient’s histopathological diagnosis to the date of death or the last valid follow-up, whichever comes first. Baseline serum NSE was measured within 7 days before or after the diagnosis. If multiple tests are available during this period, the one closest to the date of diagnosis will be chosen. Serum NSE was treated differently for multiple analytical purposes: either as a binary variable dichotomized by using the median (12.93 ng/mL) as the cut-off value to guarantee ideal statistical power, or as an ordinal variable divided by quartiles, or as a quantitative variable in its original form.

We also retrieved baseline NLR, albumin (ALB), alpha-fetoprotein (AFP), carcinoembryonic antigen (CEA), CA125, CA19-9 as control variables, in consideration that systemic nutritional status, inflammatory response, and tumor biomarkers of CRC might be influential confounding factors. These baseline serum blood indicators were also measured within 7 days prior or post cancer diagnosis. They were included into the analysis as binary variables using the following recommended cut-offs in normal individuals: 35 U/L for ALB, 8.78 ug/L for AFP, 5 ug/L for CEA, 35 kU/L for CA125, 37 kU/L for CA19-9. Moreover, extensive evidence had shown that lower levels of ALB and higher levels of CEA, CA125, and CA19-9 were significantly associated with poor prognosis for CRC patients when the above cut-off values were used (Lai et al., 2011; Ramphal et al., 2019; Li et al., 2023). In addition, considering that the blood tests were extracted 7 days before or after the time of diagnosis, which may introduce variability to test results between patients, therefore we also included the measurement time (ranges from −7 to 7 days) as an important controlling variable throughout the analysis.

Statistical analysis

Descriptive statistics were used to display and compare the general characteristics of CRAD patients. Survival curves for CRAD patients with different baseline serum NSE levels were plotted and tested by Kaplan-Meier method and log-rank test. Univariate and multivariate Cox proportional hazards (CPH) models were used to estimate crude and adjusted hazard ratios (HRs) together with 95% confidence intervals (CIs) for baseline serum NSE and the OS of CRAD. To estimate the dose-response relationship of the association, we preliminarily incorporated baseline NSE as an ordinal variable in the multivariate Cox model. Subsequently, the nonlinear relationship between baseline NSE on its original continuous form and the OS was evaluated using a restricted cubic spline (RCS) curve based on CPH model. RCS with three knots was used and the reference value was the median of NSE (Harrell, 2015). The reason to choose RCS instead of other splines is because it has previously been shown that the RCS can be used to approximate complex hazard functions in the context of time-to-event data (Rutherford, Crowther & Lambert, 2015). Variables that attained less rigorous significance (p < 0.10) in the univariate Cox model were included in the subsequent multivariate Cox model. For multivariate analysis, a two-tailed p < 0.05 was deemed statistically significant. Statistical analyses were performed in R software (Version 4.1.3; R Core Team, 2022).

Results

Characteristics of study subjects

We initially identified 3,217 patients with confirmed histopathological diagnosis of CRAD between 2013 and 2018. After careful review, 564 patients with complete information were included in the final analysis. The general characteristics of the study subjects were summarized in Table 1: most patients were male (60.28%); the average age at diagnosis was 59.15 years with a standard deviation (SD) of 11.79; 35.99% and 33.87% of patients reported smoking and alcohol drinking history; 180 (31.91%) were in stages I–II and 384 (68.09%) were in stages III–IV; the median survival time of all patients was 788.00 days (Inter-quartiles range, IQR: 923.00 days); 54.61% patients had received chemotherapies and 43.62% received curative operations.

Table 1 General characteristics of 564 CRAD patients.

Characteristics	All patients (N = 564)	Lower group (NSE < 12.93 ng/mL, N = 282)	Higher group (NSE >= 12.93 ng/mL, N = 282)	Statistic	p value	
Sex				χ2 = 1.90	0.17	
Female	224 (39.72)c	120 (53.57)c	104 (46.43)c			
Male	340 (60.28)c	162 (47.65)c	178 (52.35)c			
Age at diagnosis (Year)	59.15 (11.79)a	58.71 (11.81)a	59.60 (11.77)a	t = −0.89	0.37	
Ethnicity				χ2 = 0.07	0.79	
Minorities	64 (11.35)c	31 (48.44)c	33 (51.56)c			
Han majority	500 (88.65)c	251 (50.20)c	249 (49.80)c			
Smoking history				χ2 = 2.78	0.10	
No	361 (64.01)c	171 (47.37)c	190 (52.63)c			
Yes	203 (35.99)c	111 (54.68)c	92 (45.32)c			
Alcohol drinking history			χ2 = 2.29	0.13	
No	373 (66.13)c	178 (47.72)c	195 (52.28)c			
Yes	191 (33.87)c	104 (54.45)c	87 (45.55)c			
BMI (kg/m2)	22.49 (3.05)a	22.68 (3.07)a	22.30 (3.03)a	t = 1.47	0.14	
Chemotherapy				χ2 = 1.03	0.31	
No	256 (45.39)c	134 (52.34)c	122 (47.66)c			
Yes	308 (54.61)c	148 (48.05)c	160 (51.95)c			
Curative operation				χ2 = 4.15	<0.05	
No	318 (56.38)c	147 (46.23)c	171 (53.77)c			
Yes	246 (43.62)c	135 (54.88)c	111 (45.12)c			
Clinical stage				χ2 = 10.57	<0.01	
Stage I–II	180 (31.91)c	108 (60.00)c	72 (40.00)c			
Stage III–IV	384 (68.09)c	174 (45.31)c	210 (54.69)c			
Survival length (Day)	788.00 (923.00)b	977.00 (932.00)b	546.50 (782.25)b	Z = 6.66	<0.01	
NLR (Unit free)	2.10 (1.32)b	2.04 (1.24)b	2.17 (1.45)b	Z = −1.90	0.06	
ALB (U/L)	44.90 (5.45)b	44.90 (5.01)b	44.90 (5.63)b	Z = −0.89	0.37	
AFP (ug/L)	2.85 (1.63)b	2.78 (1.44)b	3.01 (1.88)b	Z = −1.59	0.11	
CEA (ug/L)	5.63 (19.96)b	3.81 (8.71)b	8.84 (35.93)b	Z = −5.60	<0.01	
CA125 (kU/L)	13.57 (10.93)b	12.86 (8.97)b	14.88 (12.50)b	Z = −2.87	<0.01	
CA19-9 (kU/L)	13.88 (24.09)b	12.61 (18.44)b	16.89 (43.03)b	Z = −3.43	<0.01	
NSE (ng/mL)	12.93 (4.63)b	–	–	–	–	
Notes:

a Mean (SD).

b Median (IQR).

c Frequency (Proportion).

Patients were divided into two groups by using the median of serum NSE (12.93 ng/mL): the lower group (NSE < 12.93 ng/mL) and the higher group (NSE >= 12.93 ng/mL). We compared characteristics of the two groups, and statistical tests showed the differences in curative operation, clinical stage, CEA, CA125, and CA19-9. Specifically, we found that patients with higher levels of serum NSE group tended to not have curative surgery, be in stage III–IV, and have higher CEA, CA125 and CA19-9 (both p < 0.05, Table 1). Additionally, the median survival time for patients of lower NSE level was 1.79-fold longer than patients of higher NSE level (977.00 vs 546.50 days, p < 0.01 by rank-sum test).

Baseline serum NSE and the OS of CRAD

Figure 1 displays the survival curves of the two groups of CRAD patients with different baseline serum NSE levels. OS was significantly better for patients with lower NSE level than patients with higher NSE level (p < 0.01 by log-rank test). Stratified analysis by clinical stage revealed that, for early stage CRAD patients (stage I–II), baseline NSE level was not significantly associated with OS (p = 0.85 by log-rank test), whereas for patients of advanced stage (stage III–IV), baseline NSE level was prominently related to OS (p < 0.01 by log-rank test), with a better OS observed for patients of lower baseline NSE (Fig. 2).

Figure 1 Kaplan-Meier survival curves for CRAD patients with different baseline serum NSE levels.

Figure 2 (A and B) Kaplan-Meier survival curves for CRAD patients with different baseline serum NSE levels, stratified by clinical stage.

Univariate Cox regression identified 10 potential covariates from 16 candidates: age at diagnosis, BMI, curative operation, clinical stage, NLR, ALB, CEA, CA125, CA19-9, and NSE (see in Table S1). After adjustment by multivariate Cox model, baseline serum NSE remained a significant prognostic factor: the adjusted HR for CRAD patients of higher baseline NSE was 1.82 (95% CI [1.30–2.55], p < 0.01) compared with CRAD patients of lower baseline NSE (Fig. 3A). Subsequently, we performed the multivariate Cox regression in CRAD patients in different clinical stages, and the adjusted HR of serum NSE was not statistically significant in patients with stage I–II CRAD (HR: 1.07 95% CI [0.34–3.37]; p = 0.91). However, among stage III–IV patients, the higher level of serum NSE had a 1.92-fold death hazard than lower level (p < 0.05; Fig. 3B). We further performed a sensitivity analysis in stage III-IV CRAD patients stratified by curative operation, and found the adjusted NSE-OS association was statistically significant, regardless of whether curative operation was performed (both p < 0.01; see in Fig. S1).

Figure 3 (A and B) Multivariate Cox proportional hazards model fitting results for OS of CRAD patients.

Dose-response association between baseline NSE and OS of CRAD

To test the robustness and disclose the trend of this significant NSE-OS association, we divided study subjects into four groups according to quartiles of baseline serum NSE: group 1 (NSE < 11.08 ng/mL), group 2 (11.08 <= NSE < 12.93 ng/mL), group 3 (12.93 <= NSE < 15.72 ng/mL), and group 4 (NSE >= 15.72ng/mL). By taking group 1 as the reference, multivariate Cox model disclosed that, along with the increase of baseline NSE, the death hazard of CRAD patients also increased: the adjusted HRs were 1.84 (95% CI [1.07–3.17], p < 0.05), 1.81 (95% CI [1.08–3.03], p < 0.05), and 3.30 (95% CI [2.00–5.47], p < 0.01) for group 2, group 3, and group 4, respectively (Fig. 4).

Figure 4 Quartiles of baseline NSE and the OS of CRAD patients.

Adjusted by age at diagnosis, BMI, curative operation, clinical stage, NLR, ALB, CEA, CA125, CA19-9.

We further delineate the trend of this possible dose-response relationship. RCS curve was used by including NSE in its original quantitative form. The RCS supported a prominent dose-response association in CRAD patients: along with the increase of baseline serum NSE, the adjusted HR of CRAD patients increased gradually, and the adjusted HR rose particularly steeply for baseline RCS of the interval 12.93–18.05 ng/mL (Fig. 5A). Moreover, a similar dose-response relationship for the NSE-OS association existed in stage III–IV CRAD patients, but this association was apparently absent in stage I–II patients (Figs. 5B and 5C).

Figure 5 (A–C) Adjusted HRs with 95% confidence band for the NSE-OS association of CRAD patients by using RCS.

Adjusted by age at diagnosis, BMI, curative operation, NLR, ALB, CEA, CA125, CA19-9. (A) Additionally adjusted for clinical stage.

Discussion

In the present retrospective study, as expected, we found a prominent association between baseline serum NSE and the OS of CRAD patients: after controlled for possible confounding variables, CRAD patients with higher level of baseline NSE were observed inferior OS compared with CRAD patients with lower level of baseline NSE, the elevated baseline NSE had been associated with a HR of 1.82, with apparent dose-response trend. These primary findings suggest that serum NSE at diagnosis could be a meaningful prognostic marker for CRAD patients.

To the best of our knowledge, no existing studies had exclusively investigated the association between serum NSE and survival outcomes for CRC patients. However, this positive association had been reported in other malignant tumors. For instance, Bremnes et al. (2003) revealed that increased pretreatment serum NSE was associated with worse disease-specific survival (DSS) for SCLC patients. Another prospective study of 621 NSCLC patients by Pujol et al. (2001) showed that patients with lower baseline serum NSE had better survival. In a recently published study by Yan et al. (2021), the authors found that in NSCLC patients, higher level of preoperative serum NSE predicted worse progression-free survival (PFS), with similar results in subsequent subgroup analysis of lung adenocarcinoma patients. Our study results were generally in agreement with these previous studies in supporting the detrimental role of baseline serum NSE in cancer prognosis.

Several mechanisms might be involved in the association between serum NSE and the OS of CRAD. First, NSE is a critical enzyme in aerobic glycolysis. Tumor cells with high NSE expression could accelerate glycolysis and thereby proliferate more quickly (Isgrò, Bottoni & Scatena, 2015; Vizin & Kos, 2015). Second, like SCLC tumor cells, CRAD tumor cells may secrete serum NSE, which correlates with tumor mass extension (Carney et al., 1982), therefore higher level of serum NSE may indicate aggravated tumor burden of the patients. This hypothesis had been partly justified in this study, as most patients with higher level of serum NSE were at advanced stage (Stage III–IV). However, our stratified analysis results suggest that, even controlled for tumor stage, the independent association between NSE and OS of CRAD patients was still significant. Third, it has been found that NSE was overexpressed in CRC, which implies that CRC cells could be more aggressive. In addition, NSE was obviously upregulated in metastatic colon cancer cell lines, indicating that NSE is associated with metastatic in vitro and in vivo, higher level of NSE could be more prone to metastasis (Pan et al., 2020).

Another important finding of the current study is that clinical stage presented as a prominent effect modifier in NSE-OS association, as the association was statistically significant only for advanced CRAD patients. This stage-specific NSE-OS association probably suggests that, for advanced CRAD patients, a simple blood test of serum NSE should be simultaneously considered for identifying individuals of less optimistic survival outcome. In prior studies, investigators had identified significant associations between serum NSE and clinical stage and tumor size in patients with tumors, with higher serum NSE commonly observed in patients with advanced stage and larger tumor size (Georgantzi et al., 2018; Luo et al., 2020). This might be due to elevated serum NSE occurring concurrently with malignant proliferation in cancer patients, which can directly reflect a patient’s tumor burden (Isgrò, Bottoni & Scatena, 2015). One thing to be noticed is that, although for early stage (Stage I–II) patients, we failed in observing a significant NSE-OS association, considering that only limited number of patients (N = 180) were in this group, which undoubtedly hampers statistical power, and increases the possibility of false negative inference, future studies for early stage CRAD patients of larger sample sizes are warranted.

The major findings of our study emphasize a promising role of serum NSE in prognosis of CRAD. Monitoring changes of serum NSE could be meaningful in tracing tumor progression and therapeutic response for CRAD patients. A latest study showed that for elderly CRC patients who underwent laparoscopic radical surgery, those who accepted intraoperative intravenous dexmedetomidine showed lower level of postoperative serum NSE than patients who accepted intraoperative intravenous sodium chloride (Tang et al., 2022). Dexmedetomidine might be beneficial for the proliferation and differentiation of neurons and peripheral cells in CRC patients, thereby could alleviate postoperative neurological damage (Tang et al., 2022). Another study observed that cinobufacin combined with chemotherapy can effectively decrease serum NSE and chemotherapy toxicity in NSCLC patients (Zeng et al., 2021). In addition, an animal study found that lung cancer progression was greatly inhibited by oral vanillic acid to reduce serum NSE in mice (Velli et al., 2019). Further studies on these drugs may support new treatment options for CRAD patients.

Our study is among the first to investigate the prognostic role of serum NSE in CRAD patients. Multivariate model and the subsequent dose-response analyses all reached congruent results, which further corroborates the robustness of the positive association between serum NSE and the OS of CRAD. Nonetheless, some limitations should not be ignored. First, our study was retrospective with risk of information bias. Second, patients were screened out from a single institution, so the results should be extrapolated with caution. Third, only 564 from 3,217 CRAD patients with complete data were analyzed, which may further introduce selection bias. Finally, due to data limitations, our study was unavailable to obtain information for other possible confounders or potential indicators of study interest, such as resection margin status, presence of positive regional lymph nodes, patient NSE immunohistochemical staining samples, progression-free survival outcomes, post-operative NSE measurements, and baseline genomic characteristics regarding to CRAD patients of different NSE levels, future studies should be done to address these deficiencies.

Conclusions

In conclusion, elevated serum NSE measured at diagnosis was associated with poorer OS in CRAD patients, especially for patients of advanced stage. Baseline serum NSE might be a meaningful prognostic marker for CRAD. Future studies of multiple data sources and prospective design should be done to corroborate our major findings.

Supplemental Information

Supplemental Information 1 Univariate Cox model fitting results.

Supplemental Information 2 Stratified analysis presenting the adjusted associations between baseline NSE and the OS of stage III–IV CRAD patients by curative operation.

Supplemental Information 3 Raw data.

Additional Information and Declarations

Competing Interests

Author Contributions

Human Ethics

Data Availability

The authors declare that they have no competing interests.

Junwei Peng performed the experiments, analyzed the data, prepared figures and/or tables, authored or reviewed drafts of the article, and approved the final draft.

Jie Ma performed the experiments, analyzed the data, prepared figures and/or tables, authored or reviewed drafts of the article, and approved the final draft.

Jian Lu performed the experiments, prepared figures and/or tables, and approved the final draft.

Hailiang Ran performed the experiments, prepared figures and/or tables, and approved the final draft.

Zhongqin Yuan performed the experiments, prepared figures and/or tables, and approved the final draft.

Hai Zhou performed the experiments, prepared figures and/or tables, and approved the final draft.

Yunchao Huang conceived and designed the experiments, authored or reviewed drafts of the article, and approved the final draft.

Yuanyuan Xiao conceived and designed the experiments, authored or reviewed drafts of the article, and approved the final draft.

The following information was supplied relating to ethical approvals (i.e., approving body and any reference numbers):

The Ethics Review Board of Kunming Medical University.

The following information was supplied regarding data availability:

The raw data for analysis of this manuscript is available in the Supplemental File.

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
