# Peer review of "Serum neuron-specific enolase (NSE) is associated with the overall survival of colorectal cancer: a retrospective study"

_PeerJ, doi:10.7717/peerj.18617_

## Round 0.1 · original submission · Major Revisions

Methodology section and results need to rechecked. Please provide justification for using the NSE cutoff from another cancer type and stratification by key variables. Also NSE values should be provided in table 1. Manuscript needs to be revised extensively to address comments from all 3 reviewers. .

Reviewer 1 ·

Basic reporting

In this retrospective study, Authors of the article investigated NSE in a cohort of CRC patients. In general, the manuscript has a clear message, however, some editing is still required. The following issues were found.

1. In itroduction, to show how NSE and CRC is more related, Authors should also discuss the (neuro)endocrine differentiation occuring in "simple" adenocarcinomas. This phenomenon affects not only NSE, but chromogranins A and B as well, which were investigated in a CRC by a few papers as well recently. Moreover, other neuroendocrine markers have also been investigated lately. The very short presentation of these results might give the readers a much clearer base, why this study is important for the field.

2. In Methods, Authors should give further details about the following.
2.A. NSE is not a standard tumor marker in the case of CRC. Please give some indication why such a wide range of tumor markers were investigated in the Institute of the Authors.
2.B. Please give further details about the "recommended cut-off" values. Were these investigated in relation to good - poor survival rates, etc.?
2.C. The statistical modeling details have to be extended. Specifically, the description of the restricted cubic spline modeling is very unclear. Its description is mixed with the ordinal conversion, and therefore, it is hard to understand, what exactly happened. In the RCS modeling what was the dependent (y) and indepent (x) variables, where the spline was applied exactly, why the restricted spline method was applied vs. natural splines, etc.

3. In Results, several of the observed differences are presented without p-values. Although most of them can be found in the tables and/or figures, but it should be easier for the readers if Authors either write the p-values in '()'-s, or make a reference to table / figure X.

4. Moderate English corrections are needed, either from a native speaking colleague, or from a professional langue editing service.

Experimental design

The design of the study is appropriate to investigate the presented question.
Please see my comments about the issues of Methods in the Basic section.

Validity of the findings

Reviewer suggests the following:
In the conclusion the Authors make the followinf statement: "elevated serum NSE measured at diagnosis was associated with poorer OS in CRAD patients,especially for patients of advanced stage". Did Authors consider excluding the stage I-II patients from the cohort, and analyze the data without them? How do this affect the results?

Additional comments

It is known for several (possible) biomarker that if the primary tumor is removed, the level of the investigated biomarker also reduces. This phenomenon can also confirm that the elevation was caused by the tumor itself, and also the post-operative levels of these biomarkers have better predictive value, compared to the preoperative ones. (Naturally this is only the case for those patients, who can be operatioed.) The Reviewer has the assumption, that this might be the case with NSE as well. Are there maybe postoperative measurements available, with which the Authors can prove this hypothesis?

In connection with my 1st question at the Basic section (neuroendocrine differentiation), is it feasible to investigate a small subcohort of the patients, whether different NSE IHC staining can be detected in the low and high cases? The Reviewer knows it is outside of the question of the current study, but this can be maybe investigated in a follow-up study.

·

Basic reporting

The manuscript presents a valuable investigation into the prognostic role of NSE in colorectal adenocarcinoma (CRAD), with a well-structured study design and relevant citations. However, there are several areas that need improvement. The manuscript contains grammatical errors and inconsistent terminology, especially switching between "CRAD" and "CARD." The use of an NSCLC-derived NSE cutoff lacks proper justification for CRAD, and missing data in Table 1 reduces clarity. Additionally, the interpretations of key figures, particularly regarding stage-specific differences, are too brief. Addressing these issues will improve the overall clarity and completeness.

Experimental design

The research question is relevant and addresses a meaningful knowledge gap by investigating the prognostic role of NSE in colorectal adenocarcinoma (CRAD), which is not widely studied. However, the methods, while generally sound, lack clarity in certain areas, such as the justification for using the NSE cutoff from another cancer type and stratification by key variables. Additionally, the 7-day window for measuring NSE could introduce variability that should be controlled. The study adheres to ethical standards, but improving methodological detail would enhance reproducibility.

Validity of the findings

The findings are generally sound, but the use of an unvalidated NSE cutoff and the lack of proper stratification raise concerns about the robustness of the data. The statistical analyses, such as Cox regression and restricted cubic splines, are appropriate, but without clear control for confounding factors (e.g., surgery status and disease stage), the conclusions may overstate NSE's role. The conclusions are linked to the research question but should be tempered to reflect potential limitations in the data.

Additional comments

The manuscript investigates the prognostic value of baseline serum Neuron-Specific Enolase (NSE) in colorectal adenocarcinoma (CRAD) patients. Through a retrospective study, the authors analyze the association between NSE levels and overall survival (OS), suggesting that higher baseline NSE is linked with worse OS. While the findings support NSE as a potential prognostic marker for CRAD, several methodological concerns, such as the use of an unvalidated NSE cutoff and lack of stratification by important clinical factors, raise questions about the robustness of the conclusions. Please see below for details.

Table 1

1. The authors used a cutoff for NSE (16.3 ng/ml) from a study on non-small cell lung cancer (NSCLC), but this threshold is not validated for colorectal adenocarcinoma (CRAD). The cutoff was originally derived based on the 95% specificity for a normal population in the NSCLC cohort of the Chen et al. study. Given the differences in tumor biology between NSCLC and CRAD, directly copying this threshold may not be appropriate for colorectal cancer patients. The authors should tune a cutoff specific to their own cohort, potentially using methods such as ROC curve analysis, to establish an optimal threshold for NSE that better reflects its prognostic value in CRAD.

2. The values for NSE in Table 1 are missing, despite NSE being a central marker in this study.

3. In table 1, authors reported statistical differences in key clinical parameters (curative operation, clinical stage, NLR, CEA, CA125, and CA19-9) between the low and high NSE groups. These differences suggest that patients with higher NSE levels tend to have more advanced disease and worse prognosis (as indicated by higher tumor markers like CEA). It would be beneficial if the authors could discuss these findings in greater detail.

Figure 1 and 2

The descriptions provided in the manuscript are too brief and do not adequately interpret the nuances and implications of the data. For instance, the significant association between NSE and OS in late-stage CRC but not in early-stage CRC could indicate that, in early-stage patients, NSE may not play a critical prognostic role. The manuscript should expand on this point to better explain these stage-specific differences.

Figure 3

1. Including Progression-Free Survival (PFS) alongside Overall Survival (OS) could provide additional insights into the impact of NSE on disease progression. If PFS data is available, it would enhance the robustness of the findings. However, if not available, the OS analysis alone is acceptable.

2. The inclusion of curative operation as a covariate in the Cox model requires further consideration. Surgery drastically changes patient outcomes, especially with respect to tumor burden and prognosis. Therefore, combining pre-operative and post-operative patients in the same analysis introduces confounding. It would be beneficial to stratify the analysis based on pre-op and post-op samples or conduct separate analyses for these groups. This would help clarify the independent effect of NSE on survival without conflating it with the impact of surgery. Such stratification would improve the accuracy and clinical relevance of the findings.


Figure 4 and 5

Without proper stratification or adjustment, the observed association between NSE and overall survival (OS) may not reflect a true direct relationship between NSE levels and survival in colorectal cancer (CRC) patients. Instead, this association could be confounded by other factors, such as advanced disease stage, age, or pre/post-operative status. For example, if patients with higher NSE levels tend to be in more advanced stages, and advanced stages are naturally associated with worse outcomes, then the relationship between NSE and survival could be partially or fully explained by the fact that patients with worse survival outcomes are more likely to have advanced disease, not necessarily because of high NSE alone.

Method section:

The method section states that baseline serum NSE was measured within 7 days before or after diagnosis. This time window introduces the potential for variability in NSE levels. If patients undergo any treatment, surgery, or experience disease progression within that period, their NSE levels may not accurately reflect the true baseline at diagnosis. For instance, patients measured post-surgery might have lower NSE levels due to tumor removal, while pre-surgery levels could be higher because of the full tumor burden. The authors should standardize NSE measurements before surgery or treatment, or conduct a sensitivity analysis to ensure the results are robust when only considering pre-intervention measurements.

Reviewer 3 ·

Basic reporting

Peng et al submitted a well written manuscript Investigating the relationship between baseline NSE levels and OS in CRAD patients.

Experimental design

Authors included 564 patients in this analysis and used a Univariate and multivariate Cox
159 proportional hazards models to estimate HR for baseline NSE levels and OS. Authors also performed dose dependent association and compare quantiles of baseline NSE level to OS.

Validity of the findings

Based on the review I have following questions:

1. Is the Dose-response association between baseline NSE and OS of CRAD analysis performed in late staged patients only? If not it might be worth looking at dose response association with OS in early and late stage patients separately.

2. How do the baseline genomic characteristic compare between NSE low and high cohort? if cfDNA or tissue DNA data is available.

---

## Round 0.2 · accepted · Accept

Authors have addressed all of the reviewers' comments and in my opinion the manuscript is ready for publication.

Reviewer 1 ·

Basic reporting

The manuscript improved significantly. Both its clarity and English have reached good quality.

Experimental design

I have not further questions. Authors adequately answered all my previous ones.

Validity of the findings

I have not further questions. Authors adequately answered all my previous ones.

Additional comments

I have not further questions. Authors adequately answered all my previous ones.

·

Basic reporting

no comment

Experimental design

no comment

Validity of the findings

no comment

Additional comments

The authors have addressed my concerns in the revised manuscript. Key improvements include:

Missing information in Table 1 has been incorporated, with a more detailed discussion of the clinical parameters differentiating the low and high NSE groups. More appropriate cutoff was applied.
The stage-specific differences in the NSE-OS association are now better explained, particularly for late-stage CRC.
The authors have acknowledged the limitation of not including PFS data and addressed the impact of curative operation status through a stratified analysis for late-stage patients.
Potential confounding factors and the variability of baseline NSE measurements due to the 7-day window have been discussed appropriately.